# Enhancing Arrhythmogenic Right Ventricular Cardiomyopathy Detection and Risk Stratification: Insights from Advanced Echocardiographic Techniques

**DOI:** 10.3390/diagnostics14020150

**Published:** 2024-01-09

**Authors:** Natália Olivetti, Luciana Sacilotto, Danilo Bora Moleta, Lucas Arraes de França, Lorena Squassante Capeline, Fanny Wulkan, Tan Chen Wu, Gabriele D’Arezzo Pessente, Mariana Lombardi Peres de Carvalho, Denise Tessariol Hachul, Alexandre da Costa Pereira, José E. Krieger, Mauricio Ibrahim Scanavacca, Marcelo Luiz Campos Vieira, Francisco Darrieux

**Affiliations:** 1Arrhythmia Unit, Instituto do Coração (InCor), Hospital das Clínicas HCFMUSP, Faculdade de Medicina, Universidade de São Paulo, Sao Paulo 05403-900, Brazil; lu.sacilotto@gmail.com (L.S.); tanchen.cardio@gmail.com (T.C.W.); darezzopessente@gmail.com (G.D.P.); denise.hachul@gmail.com (D.T.H.); mauricio.scanavacca@incor.usp.br (M.I.S.); 2Laboratory of Genetics and Molecular Cardiology, Instituto do Coração (InCor), Hospital das Clínicas HCFMUSP, Faculdade de Medicina, Universidade de São Paulo, Sao Paulo 05403-900, Brazil; lscapeline@gmail.com (L.S.C.); fwulkan@gmail.com (F.W.); mariana.lpcarvalho@hc.fm.usp.br (M.L.P.d.C.); acplbmpereira@gmail.com (A.d.C.P.); j.krieger@hc.fm.usp.br (J.E.K.); 3Echocardiogram Imaging Unit, Instituto do Coração (InCor), Hospital das Clínicas HCFMUSP, Faculdade de Medicina, Universidade de São Paulo, Sao Paulo 05403-900, Brazil; dbmoleta@gmail.com (D.B.M.); mluiz766@terra.com.br (M.L.C.V.); 4Echocardiogram Imaging Unit, Hospital Israelita Albert Einstein, Sao Paulo 05652-900, Brazil; lucas.franca81@gmail.com

**Keywords:** arrhythmogenic right ventricular cardiomyopathy, cardiomyopathy, echocardiogram, diagnosis, speckle-tracking echocardiography, strain, risk stratification

## Abstract

Introduction: The echocardiographic diagnosis criteria for arrhythmogenic right ventricular cardiomyopathy (ARVC) are highly specific but sensitivity is low, especially in the early stages of the disease. The role of echocardiographic strain in ARVC has not been fully elucidated, although prior studies suggest that it can improve the detection of subtle functional abnormalities. The purposes of the study were to determine whether these advanced measures of right ventricular (RV) dysfunction on echocardiogram, including RV strain, increase diagnostic value for ARVC disease detection and to evaluate the association of echocardiographic parameters with arrhythmic outcomes. Methods: The study included 28 patients from the Heart Institute of São Paulo ARVC cohort with a definite diagnosis of ARVC established according to the 2010 Task Force Criteria. All patients were submitted to ECHO’s advanced techniques including RV strain, and the parameters were compared to prior conventional visual ECHO and CMR. Results: In total, 28 patients were enrolled in order to perform ECHO’s advanced techniques. A total of 2/28 (7%) patients died due to a cardiovascular cause, 2/28 (7%) underwent heart transplantation, and 14/28 (50%) patients developed sustained ventricular arrhythmic events. Among ECHO’s parameters, RV dilatation, measured by RVDd (*p* = 0.018) and RVOT PSAX (*p* = 0.044), was significantly associated with arrhythmic outcomes. RV free wall longitudinal strain < 14.35% in absolute value was associated with arrhythmic outcomes (*p* = 0.033). Conclusion: Our data suggest that ECHO’s advanced techniques improve ARVC detection and that abnormal RV strain can be associated with arrhythmic risk stratification. Further studies are necessary to better demonstrate these findings and contribute to risk stratification in ARVC, in addition to other well-known risk markers.

## 1. Introduction

Arrhythmogenic cardiomyopathy (ACM) is a progressive heritable cardiac condition characterized by fibro-fatty replacement of the myocardium that predisposes patients to encompass a broad spectrum of phenotypes [1]. Arrhythmogenic right ventricular cardiomyopathy (ARVC) is the most classical form of ACM. ARVC predisposes to exercise-related ventricular arrhythmias, sudden death, and heart failure [2]. The pathological hallmarks of the disease, such as fibro-fatty infiltration, are present in the RV but may also occur in the left ventricle (LV) and can be segmental or patchy. The fibro-fatty scar tissue progresses from the subepicardial muscle layer towards the endocardium, ultimately resulting in transmural lesions with focal or diffuse wall thinning. This implies that the ventricular wall is weakened, especially the relatively thin, free RV wall, which may lead to typical aneurysmal dilatation [3].

Pathogenic variants in the genes encoding desmosomal proteins, which are important in cell-to-cell adhesion, play a key role in the pathogenesis of ARVC [4]. Cardiac desmosomes are composed of a group of proteins, including cadherins, armadillo proteins, and plakins, that provide mechanical connections between myocytes. However, non-desmosomal genes have also been identified [5,6]. Recent insights into the genetic etiology and pathophysiology using experimental pre-clinical models identified novel signaling cascades and cellular mechanisms toward targeted therapeutic strategies. Cardiac intercellular junctions are the central pathogenetic frameworks in ACM, and structural alterations in composition and remodeling are drivers of the disease [7]. Functional studies used spatial transcriptomics to characterize the fibro-fatty replacement in an explanted heart from an ARVC patient with a PKP2 mutation [8]. 

This disease affects young people, mainly athletes, and it is more frequent in males [4]. The current diagnosis of arrhythmogenic right ventricular cardiomyopathy (ARVC) is based on the revised international Task Force (TF) criteria proposed in the 2010 guidelines [9] and upgraded in 2020. The echocardiogram (ECHO) criteria are based on segmental wall motion abnormalities (akinesia, dyskinesia, or aneurysm/diastolic bulging) combined with right ventricle (RV) dilatation and/or RV dysfunction. [9,10]. Although these ECHO criteria are highly specific, their sensitivity is low, especially in the early stages of the disease [11]. ECHO is the most widely available imaging method and is considered more suitable for use in patients with defibrillators compared to cardiac magnetic resonance (CMR) [11,12]. However, the evaluation of the RV function and volume by conventional visual echocardiography can be challenging because of its retrosternal position and its complex geometry [11]. 

The analysis of left ventricular systolic function using the measurement of ejection fraction employing various imaging techniques remained for many decades the “gold standard” method for determining left ventricular myocardial systolic function. The analysis of the systolic function of the right ventricle was determined for a long period by echocardiographic methods such as FAC and TAPSE due to the special anatomical conformation of the right ventricle. However, advanced techniques for measuring left ventricular myocardial function, such as ventricular strain analysis, allow for the observation of left ventricular myocardial dysfunction even in the presence of a preserved ejection fraction [13,14,15,16,17,18,19,20]. Currently, the two-dimensional longitudinal strain of the right ventricle also allows for the analysis of the myocardial systolic function of the right ventricle and can be used in several clinical situations [17,19]. Right ventricular (RV) strain better represents RV systolic function than longitudinal measurements such as tricuspid annular plane systolic excursion (TAPSE) and s’ velocity, which primarily assess the movement of the lateral tricuspid annulus [20,21]. RV strain is a more reproducible and better discriminating prognostic tool than conventional parameters. It is less angle-dependent and closer to a more global assessment of RV function than any of the longitudinal parameters [20,21]. 

Since the description of the disease made by Marcus et al. in 1982 [2], the diagnostic criteria for right ventricular arrhythmogenic cardiomyopathy have been modified due to the pendulum between sensitivity and specificity, seeking to bring the greater possibility of application and investigation in the face of clinical suspicion [10]. In this sense, the employment of echocardiographic strain may bring some increase to the diagnostic investigation in the suspicion of right ventricular arrhythmogenic cardiomyopathy.

Two-dimensional (2D) strain using speckle-tracking echocardiography (STE) can measure myocardial deformation by acoustic markers frame by frame [11]. STE imaging enables the quantification of regional myocardial deformation [22,23], detecting subtle functional abnormalities, even in the absence of structural abnormalities, by conventional visual ECHO [24], including asymptomatic mutation carriers [25,26]. RV mechanical dispersion, assessed by strain, reflects electrical dispersion in ARVC [22]. The role of speckle-tracking echocardiography in ARVC has not been fully explored yet, especially when compared to conventional ECHO parameters. The purposes of the study were to describe RV strain measurement to enhance ARVC detection and to evaluate the association between echocardiographic parameters and arrhythmic outcomes.

## 2. Materials and Methods

A total of 28 patients with ARVC diagnosis at *Instituto do Coração (InCor), Hospital das Clínicas HCFMUSP* were enrolled to perform the ECHO with advanced techniques, including STE. Overall, the patients had undergone conventional visual ECHO and CMR at the beginning of the follow-up. The ECHO parameters with advanced techniques were compared to prior conventional visual ECHO and CMR. 

Life-threatening arrhythmic events (LTAEs) were considered any of the following arrhythmic events after enrolment: sustained ventricular tachycardia (VT), aborted sudden cardiac death, appropriate therapies from implantable cardioverter–defibrillators (ICDs), and sudden cardiac death (SCD). 

The inclusion criteria were patients who met the diagnostic criteria for arrhythmogenic right ventricular cardiomyopathy (according to the TFC 2010) and signed the informed consent form. The exclusion criteria were patients diagnosed with other cardiomyopathies, such as Chagas cardiomyopathy, sarcoidosis, and ischemic cardiomyopathy.

DNA from peripheral blood samples was purified using the QIAamp DNA Blood Mini Kit (Qiagen, Hilden, Germany), following the manufacturer’s instructions. A gene panel was customized for 160 genes within a cardiomyopathies panel, including the key genes (*PKP2*, *DSC2*, *DSG2*, *DSP*, *JUP*, *CTNNA3*, *DES*, *TMEM43*, *PLN*, *FLNC*, and *RBM20*) associated with ARVC, utilizing the Design Studio Assay Design Tool (Illumina, San Diego, CA, USA). Customized panel sequencing was conducted using the Nextera DNA Library. Preparation Kit on an Illumina MiSeq System (Illumina, San Diego, CA, USA). FASTQ files were analyzed on CLC Genomics Workbench 9 for variant calling and annotation. The sequence reads generated were mapped to the GRCh37 reference genome. Variant interpretation was conducted in accordance with the American College of Medical Genetics (ACMG) criteria [27].

The 2D STE transthoracic ECHO was acquired by echocardiogram cardiologists. The operators were blinded to the clinical data during analysis. Images were captured using an ultrasound machine, GE Vivid E95, and Philips Epiq 7C in the left lateral decubitus position. RV dimensions were evaluated by diameters measurement in apical four-chamber focused view for proximal right ventricular outflow tract (RVOT) parasternal long axis (PLAX) and parasternal short axis (PSAX) views during end-diastole. RV function was evaluated with fractional area change (FAC), tricuspid annular plane systolic excursion (TAPSE), peak systolic velocity (s’ wave) of the tricuspid annulus by pulsed-wave tissue Doppler, RV global longitudinal strain (RVGLS), and RV free wall longitudinal strain (RVFWLS). The left ventricular ejection fraction (LVEF) was calculated according to the Simpson method. Typical structural changes, such as increased reflectivity of the moderator band and prominent trabeculations of the RV apex, were described in the report when detected. Software (version 6.2.0.0) analyses were performed by IntelliSpace Cardiovascular. 

### Statistical Analysis

The SPSS statistical program version 21 (IBM Corporation, Armonk, NY, USA) and R version 4.0 (R Core Team, 2021) were used for the statistical analyses. Continuous data are presented as the mean ± standard deviation (SD) or median {IQ range}. Fisher’s exact test and the chi-squared test were used for categorical data comparison and the Wilcoxon test or the *T*-test were used for the comparison of continuous variables. Receiver operating characteristic (ROC) analysis was performed to determine the optimal cutoff values of the strain parameters and to determine the sensitivity and specificity of each variable for detecting arrhythmic outcomes. A *p*-value < 0.05 was considered significant.

## 3. Results

### 3.1. Patient Population

The clinical data are summarized in Table 1. The mean age was 33 years, 27/28 (96%) were male, and 25/28 (89%) were probands. Physical training history was reported as intense work-related activity or regular sports in 17/28 (61%). The most frequent first clinical presentation was palpitation in 18/28 (64%) followed by syncope in 6/28 (21%) and dyspnea in 2/28 (7%). Sustained ventricular tachycardia occurred in 18/28 (64%) patients and non-sustained ventricular tachycardia (NSVT) occurred in 23/28 (82%) patients. Aborted sudden death was the first manifestation in 2/28 (7%) patients. We performed genetic testing on all 25 probands. A ‘pathogenic’ or ‘likely-pathogenic’ variant was found in 16/25 (64%) patients while 9/25 (36%) patients carried a ‘variant of uncertain significance’ (VUS). The remaining 3/28 patients were family members, with positive phenotypic expression, and were also carriers of a pathogenic or likely pathogenic desmosome variant. 

All variants in the *PKP2, DSC2,* and *DSG2* of this cohort were heterozygous. We did not have any recessive arrhythmogenic cardiomyopathy in the patients of this cohort. Two probands from different families had the same *PKP2* variant: c.2446-2A>C. Detailed information on the specific pathogenic or likely pathogenic variants, including the ACMG criteria punctuation for each variant, is available in Table 2. 

### 3.2. Advanced Speckle-Tracking ECHO Analysis

Regarding RV diameter, the majority of patients presented RV dilatation, 26/28 (93%), with RVOT PLAX > 32 mm and 24/28 (86%) with RVOT PSAX > 36 mm. Functional parameters demonstrated that the majority of patients had RV dysfunction: 22/28 (79%) patients presented RV FAC ≤ 33%, and for 6/28 (21%), the FAC measure was between 34 and 40%. The left atrial and ventricular diameter were normal in most patients, and the left ventricular diameter was dilated in 2/28 (7%) patients. The left ventricular function was normal in most patients, with 4/28 (14%) patients presenting LVEF < 50%. Furthermore, 21/28 (75%) of patients presented RV segmental abnormalities. The main ECHO parameters are shown in Table 3 and examples are demonstrated in Figure 1. 

### 3.3. Comparison of Different Methods to Detect ARVC 

All patients were in sinus rhythm either during standard ECHO or STE. No technical difficulty was reported. During the conventional visual ECHO, no patient had minor criteria and 10/28 had major criteria. The remaining did not fulfill the ECHO diagnosis according to TF 2010. The CMR-based imaging detected 19/28 (67%) fulfilling major or minor criteria for ARVC and the remaining 9 did not fulfill the CMR diagnosis criteria according to TF 2010. The ECHO’s advanced STE analyses by RVFWLS detected RV abnormalities in 26/28 (92%), and only two patients had a normal strain.

### 3.4. Arrhythmic Outcomes and the Prediction of STE Parameters in ARVC

The mean follow-up time was 5.6 ± 1.1 years. During follow-up, 2/28 (7%) patients died due to a cardiovascular cause and 2/28 (7%) underwent heart transplantation.

An arrhythmic outcome occurred in 14/28 (50%) patients; 14/28 (50%) patients developed sustained VT and 2/28 (7%) had an electrical storm. 

The right ventricle dimensional parameters demonstrated that RV dilatation was associated with arrhythmic outcomes: RVDd was 39.00 mm vs. 35.00 mm (*p* = 0.018) and RVOT PSAX was 39.00 mm vs. 36.50 mm (*p* = 0.044). The association between STE parameters and arrhythmic outcomes is shown in Table 4 and Figure 2. 

A receiver operating characteristic (ROC) analysis was performed to determine the best cut-off values of RVFWL-S for differentiating patients with arrhythmic outcomes from those without at any time within 5 years (Figure 3). The values with lower strain (<−14.35 in absolute number) were associated with arrhythmic outcomes. (Table 5).

## 4. Discussion

Our study demonstrated a high prevalence of RV dilation and RV dysfunction and a potential diagnostic yield when performing ECHO with advanced techniques, including STE. Of note, all patients presented normal dimensions and function parameters for the left ventricle. These findings demonstrated a predominant RV structural disease phenotype and normal LV structural parameters, as expected in a cohort selected by TF 2010 and with a predominant *PKP2* desmosomal genotype, consistent with the molecular diagnosis from the genetic test [28,29]. 

Right ventricular systolic echocardiographic functional analysis has improved over the last decades. The current morpho-functional ventricular criterion is based on segmental wall motion abnormalities and RV dysfunction assessed through RV FAC analyses [9,29], which was impaired in all patients in this cohort, in addition to segmental dysfunction or aneurysms. 

Measurements of greater sensitivity for ARVC were those performed in the parasternal window, which has been falling out of use in many echocardiography services. The view of the right ventricle in echocardiography is limited by spatial resolution; thus, emerging techniques for the evaluation of the RV in ARVC may play a role [30]. Strain has emerged as a noninvasive and objective marker of myocardial contractility, offering valuable information into regional and global myocardial systolic function [30]. It is noteworthy to emphasize that fully automated analysis of echocardiography images offers a swift and reproducible assessment of left ventricular longitudinal strain in comparison with visual estimation and manual tracing in preliminary analyses [31]. Future strategies employing artificial intelligence through automation software for strain determination could potentially be integrated into ARVC clinical practice, leading to time and cost savings and ultimately reducing the dependence on specialized centers for ARVC diagnoses [32]. 

Moreover, prior studies have demonstrated that RVFWLS is reduced in patients with ARVC compared with healthy controls [23,33]. The majority of patients with ARVC in our cohort presented reduced RVFWLS values, in line with prior ARVC studies [12,23,33]. Additionally, STE deformation imaging has been demonstrated as being capable of identifying relatives who are at low risk of disease progression, potentially enhancing the follow-up strategy in ARVC family screening protocols, which was not tested in our cohort with patients with established disease [34]. 

Recently, the LV global longitudinal strain was included in the 2020 International Criteria for the Diagnosis of Arrhythmogenic Left Ventricular Cardiomyopathy (ALVC) as a criterion of morpho-functional ventricular abnormalities [10] and was also integrated into the Padua criteria [29]. Thus, we hypothesize that RV strain analysis could also be added to ARVC imaging parameters to increase the current diagnostic tools for ARVC detection.

Half of the patients presented arrhythmic outcomes (LTAEs) during the follow-up period regardless of the initial presentation. Among several ECHO parameters, the RV dilatation parameters (RVDd mid and RVOT PSAX) were significantly associated with arrhythmic outcomes (LTAEs) in our cohort, which is in line with previous reports on the ARVC phenotype also demonstrating that RV dilatation is associated with arrhythmic outcomes [35,36]. Lie et al. found an association between several echocardiographic measures and arrhythmic outcomes, including RV dilatation, RV function, and left ventricular function [37]. In our cohort, the majority of patients presented normal left ventricular size and normal left ventricular function; thus, this could justify that LVEF was not associated with the outcomes. 

The RV abnormal strain is significantly associated with the risk of the structural progression of RV dilatation and dysfunction [12]. In ARVC, the greater the ventricular dysfunction, the higher the risk of adverse outcomes [38]. Hosseini et al. concluded that RV global longitudinal strain was lower in patients reaching mortality and arrhythmic endpoints [35]. In our cohort, lower STE was also associated with arrhythmic outcomes. Our study reinforces the need for further exploration of advanced ECHO-driven techniques in ARVC, either to improve the detection of RV abnormalities or to enhance risk assessments in this challenging disease. We believe that in the future, the anatomical, geometric, functional, and segmental analysis of ventricles and atria could take into account automatic evaluations based on deep learning algorithms, multi-level semantic adaptation, and temporal consistency models, bringing less interobserver variation in diverse analyses and leading to shorter study times [39,40,41,42,43].

## 5. Study Limitations 

This study has several limitations. First, it involves a single referral center, so there is potential referral bias with overestimated severity results in a cohort of patients with a more severe phenotype. Second, we reported data from a relatively small population. Although ARVC is a rare disease, the sample size of the cohort was relatively modest, which might influence the generalizability of our results. A larger and more diverse participant pool would enhance the external validity of our conclusions. Third, specificity was not predicted to be studied because a control group was not analyzed. Long-term reassessments with second analyses of echocardiograms over time would be particularly beneficial in capturing potential changes in echocardiographic measurements and arrhythmic outcomes that may not be immediately evident. Echocardiographic measurements are subject to variability in interpretation, and interobserver and intraobserver variability may influence the accuracy of the results. Lastly, the absence of certain clinical covariates in our analysis may limit the depth of interpretation of the results. Despite the limitations above, our study recognizes that ECHO is an imaging tool of important value and that these advanced acquisition data provide information on ECHO’s usefulness in identifying ARVC abnormalities that could improve ARVC detection.

## 6. Conclusions

Our data indicate that the application of advanced techniques in echocardiography can detect RV abnormalities in ARVC. Notably, abnormal RV strain emerged as a valuable contributor to arrhythmic risk stratification. In summary, the study highlights the nuanced characteristics of RV abnormalities in ARVC and suggests potential windows of opportunity for advancing diagnostic tools and risk assessment strategies in the future. While our results provide crucial insights, further studies are necessary to better demonstrate these findings. Such endeavors will not only strengthen our understanding of ARVC but also contribute to the refinement of diagnostic approaches and risk stratification, complementing the existing established risk predictors.

## Figures and Tables

**Figure 1 diagnostics-14-00150-f001:**
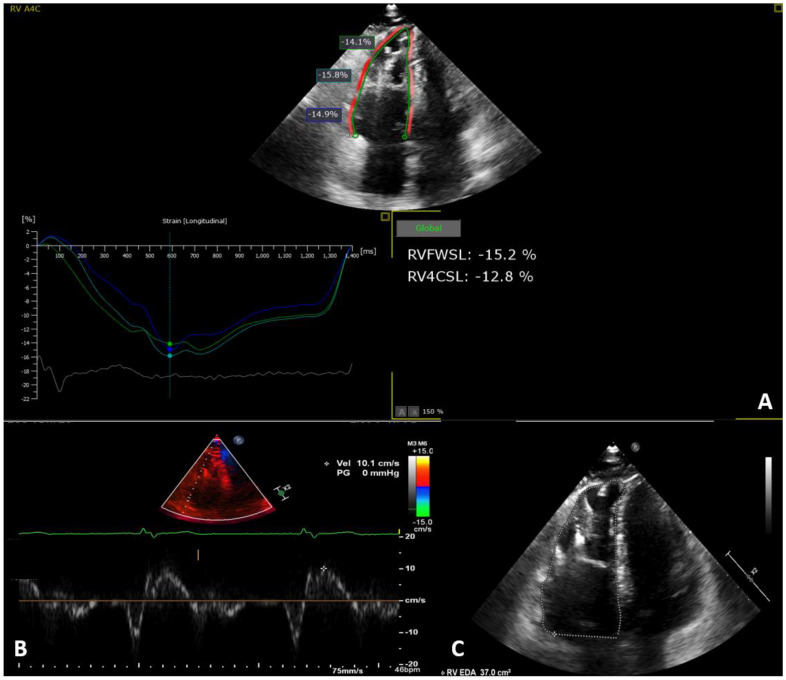
Representative image of right ventricular echocardiographic parameter measurement. (**A**) Apical four-chamber right ventricle focused view with measures of longitudinal strain systolic peak values. Right ventricular free wall longitudinal strain and right ventricular global longitudinal strain demonstrating RV systolic dysfunction: −15.2% and −12.8% values, respectively. (**B**) Pulsed-wave tissue Doppler echocardiography showing the RV peak systolic velocity (s’ wave) of the tricuspid annulus in 10.1 cm/s. (**C**) Apical four-chamber right ventricle focused view with measures of RV diastolic area change, RV systolic area change, and fractional area change estimated at 37 cm^2^, 29.3 cm^2^, and 20.8%, respectively. *RVFWSL: right ventricular free wall longitudinal strain. RV4CSL: right ventricular four-chamber longitudinal strain*.

**Figure 2 diagnostics-14-00150-f002:**
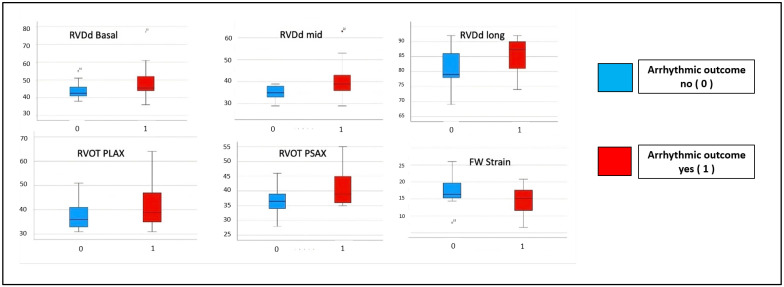
The echocardiographic measurements reveal variations in values based on arrhythmic outcomes, indicating a discernible trend toward increased dilation and dysfunction in patients experiencing arrhythmic events. In blue: patients without arrhythmic outcomes at any time within 5 years. In red: patients with arrhythmic outcomes. *PLAX*: parasternal long axis; *PSAX*: parasternal short axis; *RVDd*: right ventricular end-diastolic diameter; *RVDs*: right ventricular end-systolic diameter; *RV*: right ventricle; *RVOT*: right ventricular outflow tract.

**Figure 3 diagnostics-14-00150-f003:**
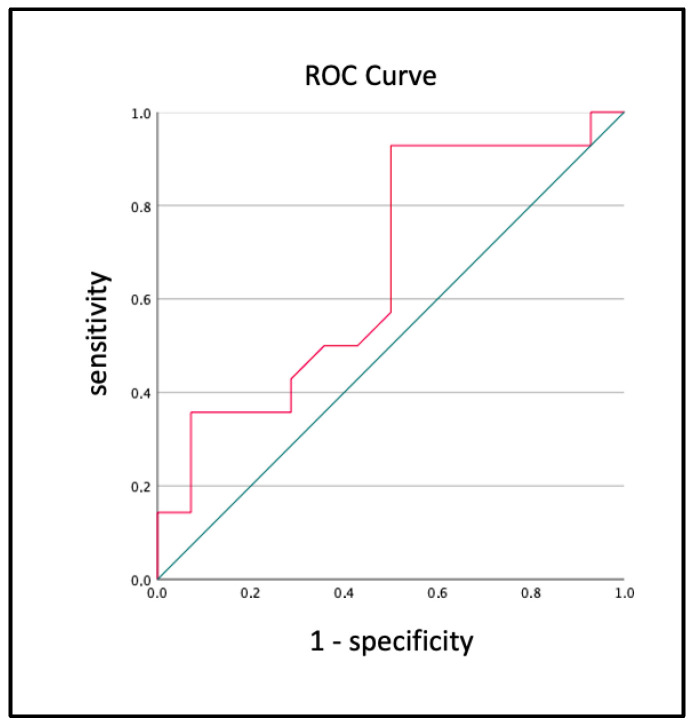
ROC curve for the strain RVFWL-S parameters.

**Table 1 diagnostics-14-00150-t001:** Baseline clinical characteristics.

Mean age (min, max); years	33 (13, 57)
Male (%)	27/28 (96)
Proband (%)	25/28 (89)
First Clinical Presentation n (%)	
- Palpitations	18/28 (64)
- Syncope	6/28 (21)
- Dyspnea	2/28 (7)
Arrhythmias, n (%)	
- VT	18/28 (64)
- NSVT	23/28 (82)
- Aborted sudden death	2/28 (7)
Genetic Test, n/N (%)	
- Pathogenic (or likely)	16/25 (64)
PKP2 nonsense	12/16 (75)
DSC2 nonsense	3/16 (19)
DSG2 nonsense	1/16 (6)

The patient cohort is composed of young patients, mostly men, and those with an arrhythmic clinic phenotype. NSVT: non-sustained ventricular tachycardia; VT: sustained ventricular tachycardia.

**Table 2 diagnostics-14-00150-t002:** Detailed information on the specific pathogenic or likely pathogenic variants.

	Pathogenic or Likely Pathogenic Variants			
Gene	Codon	Protein	Class	ACMG Criteria
*PKP2*				
NM_001005242.3	c.148_151del	p.Thr50Serfs *	4	PVS1, PM2, PS3
NM_001005242.3	c.368G>A	p.Trp123 *	5	PVS1, PM2, PP1
NM_001005242.3	c.775dupG	p.Glu259fs *	4	PVS1, PM2
NM_001005242.3	c.1170+2T>A	p.(?)	5	PVS1, PM2, PP3, PS4, PP1
NM_001005242.3	c.1643delG	p.Gly548fs *	4	PVS1, PM2
NM_004572.3	c.1951C>T	p.Arg651 *	5	PVS1, PM2, PS4, PP1
NM_004572.3	c.2062T>C	p.Ser688Profs *	5	PVS1, PM2, PS4, PP1
NM_001005242.3	c.2146-1G>C	Splice acceptor	5	PVS1, PM2, PP3, PS4
NM_001005242.3	c.2357+1G>A	p.(?)	5	PVS1, PM2, PP3, PS4
NM_001005242.3	c.2446-2A>C	p.(?)	5	PVS1, PM2, PP3,
NM_004572.3	Deletion (exon 4)	p.(?)	4	PVS1, PM2
*DSC2*				
NM_024422.6	c.923C>G	p.Ser308fs *	4	PVS1, PM2
NM_024422.6	c.929delinsTT	p.Gln310fs *	4	PVS1, PM2
NM_024422.6	c.1053_1059del	p.His351Glnfs *	4	PVS1, PM2
*DSG2*				
NM_001943.5	c.523+1G>C	p.(?)	5	PVS1, PM2, PP3, PS4

Pathogenic or likely pathogenic variants demonstrating codon and protein alterations and the ACMG classification. The pathogenic variants were predominant in the PKP2 gene. ACMG: American College of Medical Genetics.

**Table 3 diagnostics-14-00150-t003:** ECHO parameters in ARVC.

RV Diameter, Volume, and Function	
	mean ± DP or median, IQR
- RV FAC (%)	27 ± 0.08
- RVDd basal (mm)	45.00 (41.00, 48.75)
- RVDd mid (mm)	37.00 (33.75, 39.25)
- RVDd longitudinal (mm)	83.14 ± 6.64
- RVOT (PLAX) (mm)	37.50 (34.75, 41.25)
- RVOT (PSAX) (mm)	37.471 (35.75, 40.00)
- TAPSE (mm)	18.07 ± 4.38
- RVFWLS,	−15.86 ± 4.25
**LV diameter, volume, and function**	
	mean ± DP or median, IQR
- LAD (mm), mean ± DP	35.96 ± 4.64
- LAD vol (mL/m)	28.90 (26.00, 31.70)
- Septum (mm)	8.85 (7.00–11.0)
- Posterior Wall (mm)	8.41 (7.00–10.0)
- LVDd (mm), mean ± DP	47.86 ± 6.46
- LVDs (mm)	34.06 (24.0–59.0)
- LVEF (%)	0.60 (0.47, 0.66)

Echocardiogram parameters according to the right and left ventricles. The cohort has right ventricular dilation and dysfunction predominance. Data are expressed as the mean ± SD or median (IQR). FAC: right ventricular fractional area change; RVFWLS: right ventricular free wall longitudinal strain; LAD: left atrial dimension; LVDd: left ventricular end-diastolic diameter; LVEF: left ventricular ejection.

**Table 4 diagnostics-14-00150-t004:** Arrhythmic outcomes and STE parameters.

RV Diameter, Volume, and Function			
	Arrhythmia (Yes)	Arrhythmia (No)	*p*-Value *
RV FAC (%)	0.25 ± 0.09	0.30 ± 0.08	0.159
RVDd basal (mm)	45.50 (44.25, 52.00)	42.50 (41.00, 45.75)	0.075
RVDd mid (mm)	39.00 (36.25, 42.50)	35.00 (33.00, 37.75)	0.018
RVDd longitudinal (mm)	85.36 ± 5.79	80.93 ± 6.90	0.077
RVOT (PLAX) (mm)	39.00 (35.50, 45.50)	36.00 (33.00, 40.75)	0.196
RVOT (PSAX) (mm)	39.00 (36.50, 44.75)	36.50 (34.00, 39.00)	0.044
TAPSE (mm)	16.50 ± 3.94	19.64 ± 4.36	0.055
RVFWL Strain	−14.48 ± 3.85	−17.24 ± 4.32	0.086
LAD (mm)	34.64 ± 4.09	37.29 ± 4.92	0.134
LAD vol (mm)	28.50 (25.25, 30.05)	29.50 (26.70, 32.30)	0.382
LVDd (mm)	45.79 ± 7.20	49.93 ± 5.06	0.091
LVDs (mm)	31.64 ± 5.93	33.29 ± 4.63	0.421
LVEF (%)	0.59 (0.56, 0.67)	0.60 (0.57, 0.66)	0.611

Right and left atrial and ventricle parameters according to arrhythmic outcomes: right ventricle dilation parameters are associated with arrhythmic outcomes. FAC: right ventricular fractional area change; RVFWLS: right ventricular free wall 2D longitudinal strain; LAD: left atrial dimension; LVDd: left ventricular end-diastolic diameter; LVEF: left ventricular ejection fraction; PLAX: parasternal long axis; PSAX: parasternal short axis; RVDd: right ventricular end-diastolic diameter; RVDs: right ventricular end-systolic diameter; RV: right ventricle; RVOT: right ventricular outflow tract; TAPSE, tricuspid annular plane systolic excursion.* The Wilcoxon test was applied for non-normal variables and the *t*-test was applied for normal variables.

**Table 5 diagnostics-14-00150-t005:** RVFWL strain in arrhythmic outcomes.

	Arrhythmic Outcomes (Yes)	Arrhythmic Outcomes (No)	*p*-Value
Strain > −14.35% (or <14.35% in absolute value)	7	1	0.033 *
Strain ≤ −14.35% (or ≥14.35% in absolute value)	7	13	
Total	14	14	

RVFWL: right ventricular free wall longitudinal. * chi-squared test.

## Data Availability

Data are not available online due to ethical considerations and confidentiality agreements of the institutional review board. We can make it available upon individual request to the corresponding authors.

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
