# Peer review of "Enhancing Arrhythmogenic Right Ventricular Cardiomyopathy Detection and Risk Stratification: Insights from Advanced Echocardiographic Techniques"

_diagnostics, 2024, doi:10.3390/diagnostics14020150_

Round 1
Reviewer 1 Report
Comments and Suggestions for Authors
In the brief report 'Enhancing Arrhythmogenic Right Ventricular Cardiomyopathy Detection and Risk Stratification: Insights from Advanced Echocardiographic Techniques' submitted by Olivetti et al., the authors analysed 28 patients with ARVC by echocardiography.
The manuscript is interesting but needs some extensions and corrections:
1.) Line 42: The authors should also explain that fibro-fatty replacement is the underlying reason for ARVC. It addition, I would mention also the right ventricle is affected at a later timepoint. This fact I would also discuss or explain. For example, the group of Eva van Rooij has used spatial transcriptomics to characterize the fibro-fatty replacement in an explanted heart from an ARVC patient with a PKP2 mutation (Boogerd CJ et al. Cardiovasc RES 2023). I would discuss this landmark paper in this context.
2.) I think the genetic background of ARVC needs some more explanations or you can introduce in line 45 a good new review article about ARVC. For example, the article 'Insights into Genetics and Pathophysiology of Arrhythmogenic Cardiomyopathy' would be a good starting point. ARVC can be also caused by non-desmosomal genes like DES (Protonotarious A, Can J Cardiol 2021) or ILK (Brodehl A, Trans Res 2019). This should be also introduced (Line 45)
3.) Who is the manufacturer of the DNA extraction kit? (Line 94).
4.) Please list some more details to the sequencing analysis. Which sequencer, which library preparation kit? Which company was involved?)
5.) All gene names should be written in Italics.
6.) Line 96: FLNC not FLC!
7.)Table 1: Please prepare a detailed table with the specific identified mutation.
8.) Are the nonsense variants in DSC2 and DSG2 heterozygous or homozygous? (Table 1). Recently, is was shown that also recessive DSC2 and DSG2 mutations contribute to ARVC/ACM. See 'A homozygous DSC2 deletion associated with arrhythmogenic cardiomyopathy is caused by uniparental isodisomy' and 'Hemi- and homozygous loss-of-function Mutations in DSG2 (Desmoglein-2) cause recessive arrhythmogenic cardiomyopathy with an early onset'. This should be explained and discussed.
9.) Could you please insert a paragraph about the statistical analysis, which you used for your data analysis in the material and methods section?
Nevertheless, this manuscript is interesting to many clinical and basic scientist. However, some points can be explained in a better way and some extensions are necessary. Therefore, I suggest a major revision.
Comments on the Quality of English LanguageA native English speaking editor should double check this manuscript.
Author Response
Response to Review 1:
We sincerely appreciate the time and effort you dedicated to reviewing our manuscript. Your constructive feedback and improvements have been invaluable to us. Thank you for your time and consideration.
Our point-by-point response are below:
1- Question: Line 42: The authors should also explain that fibro-fatty replacement is the underlying reason for ARVC. In addition, I would mention also the right ventricle is affected at a later timepoint. This fact I would also discuss or explain. For example, the group of Eva van Rooij has used spatial transcriptomics to characterize the fibro-fatty replacement in an explanted heart from an ARVC patient with a PKP2 mutation (Boogerd CJ et al. Cardiovasc RES 2023). I would discuss this landmark paper in this context.
Response: We appreciate and agree with the pertinent comments. We added the requested sentence and the suggested reference (PMID: 35576477) as follows:
"Arrhythmogenic cardiomyopathy (ACM) is a progressive heritable cardiac condition characterized by fibro-fatty replacement of the myocardium that predisposes patients to encompassing a broad spectrum of phenotypes. Arrhythmogenic right ventricular cardiomyopathy (ARVC) is the most classical form of ACM that predisposes to exercise-related ventricular arrhythmias, sudden death and heart failure [1]. The pathological hallmarks of the disease are usually distinctly present in the RV but may also occur in the left ventricle (LV), and can be segmental or patchy. The fibro-fatty scar tissue progresses from the subepicardial muscle layer towards the endocardium, ultimately resulting in transmural lesions with focal or diffuse wall thinning. This implies the ventricular wall is weakened, especially the relatively thin, free RV wall, which may lead to typical aneurysmal dilatation. Pathogenic variants in the genes encoding desmosomal proteins, which are important in cell-to-cell adhesion, play a key role in the pathogenesis of ARVC [2]. Cardiac desmosomes are composed of a group of proteins, including cadherins, armadillo proteins, and plakins, that provide mechanical connections between myocytes. However, non-desmosomal genes have also been identified [5,6]. Recent insights into the genetic etiology and pathophysiology using experimental pre-clinical models identified novel signaling cascades and cellular mechanisms towards targeted therapeutic strategies. Cardiac intercellular junctions are the central pathogenetic frameworks in ACM, and structural alterations in composition and remodeling are drivers of the disease [7]. Functional studies used spatial transcriptomics to characterize the fibro-fatty replacement in an explanted heart from an ARVC patient with a PKP2 mutation [8]."
2- Question: I think the genetic background of ARVC needs some more explanations or you can introduce in line 45 a good new review article about ARVC. For example, the article 'Insights into Genetics and Pathophysiology of Arrhythmogenic Cardiomyopathy' would be a good starting point. ARVC can be also caused by non-desmosomal genes like DES (Protonotarious A, Can J Cardiol 2021) or ILK (Brodehl A, Trans Res 2019). This should be also introduced (Line 45)
Response: We appreciate your constructive feedback. Indeed, the main purpose was to clarify that our research was primarily focused on clinical diagnosis using imaging methods. As such, we deliberately limited our discussion on genetics to maintain a concise and focused narrative aligned with the primary objectives of our work. As requested, the genetic backgrounds and references was added as follows:
"Pathogenic variants in the genes encoding desmosomal proteins, which are important in cell-to-cell adhesion, play a key role in the pathogenesis of ARVC [2]. Cardiac desmosomes are composed of a group of proteins , including cadherins, armadillo proteins, and plakins, that provide mechanical connections between myocytes. However, non-desmosomal genes have also been identified [5,6]. Recent insights into the genetic etiology and pathophysiology using experimental pre-clinical models identified novel signaling cascades and cellular mechanisms towards targeted therapeutic strategies. Cardiac intercellular junctions are the central pathogenetic frameworks in ACM, and structural alterations in composition and remodeling are drivers of the disease [7]. Functional studies used spatial transcriptomics to characterize the fibro-fatty replacement in an explanted heart from an ARVC patient with a PKP2 mutation [8]."
3- Question: Who is the manufacturer of the DNA extraction kit? (Line 94).
Response: It was utilized the QIAamp DNA Blood Mini Kit (Qiagen, Hilden, Germany), following the manufacturer instructions. We added a sentence specifying this issue:
“DNA from peripheral blood samples was purified using the QIAamp DNA Blood Mini Kit (Qiagen, Hilden, Germany), following the manufacturer instructions.”
4- Question: Please list some more details to the sequencing analysis. Which sequencer, which library preparation kit? Which company was involved?)
Response: The sequencer used was next-generation sequencing (NGS) and the library preparation kit was Nextera DNA Library. We added a paragraph with details on the sequencing analysis, as follows:
“A gene panel was customized for 160 genes within a cardiomyopathies panel, including the key genes (PKP2, DSC2, DSG2, DSP, JUP, CTNNA3, DES, TMEM43, PLN, FLNC, RBM20) associated with ARVC, utilizing the Design Studio Assay Design Tool (Illumina, San Diego, CA). Customized panel sequencing was conducted using the Nextera DNA Library. Preparation Kit on an Illumina MiSeq System (Illumina, San Diego, CA). FASTQ files were analyzed on CLC Genomics Workbench 9 for variant calling and annotation. The sequence reads generated were mapped to the GRCh37 reference genome. Variant interpretation was conducted in accordance with the American College of Medical Genetics (ACMG) criteria [25].”
5- Question: All gene names should be written in Italics.
Response: Thanks for the advice. We wrote all genes in italics.
6- Question: Line 96: FLNC not FLC!
Response: Apologies for the typing error (FLC instead of FLNC). The mistake has been corrected in the line, and it now reads correctly (FLNC). Thank you for pointing out this error.
7- Question: Table 1: Please prepare a detailed table with the specific identified mutation.
Response: We added a table with detailed specific variants. Two probands from different families had the same PKP2 variant c.2446-2A>C.
|
Pathogenic or Likely Pathogenic Variants |
|
|
|
|||||
|
Gene |
Codon |
Protein |
Class |
ACMG criteria |
||||
|
PKP2 |
|
|
|
|
||||
|
NM_001005242.3 |
c.148_151del |
p.Thr50Serfs* |
4 |
PVS1, PM2, PS3 |
||||
|
NM_001005242.3 |
c.368G>A |
p.Trp123* |
5 |
PVS1, PM2, PP1 |
||||
|
NM_001005242.3 |
c.775dupG |
p.Glu259fs* |
4 |
PVS1, PM2 |
||||
|
NM_001005242.3 |
c.1170+2T>A |
p.(?) |
5 |
PVS1, PM2, PP3, PS4, PP1 |
||||
|
NM_001005242.3 |
c.1643delG |
p.Gly548fs* |
4 |
PVS1, PM2 |
||||
|
NM_004572.3 |
c.1951C>T |
p.Arg651* |
5 |
PVS1, PM2, PS4, PP1 |
||||
|
NM_004572.3 |
c.2062T>C |
p.Ser688Profs* |
5 |
PVS1, PM2, PS4, PP1 |
||||
|
NM_001005242.3 |
c.2146-1G>C |
Splice acceptor |
5 |
PVS1, PM2, PP3, PS4 |
||||
|
NM_001005242.3 |
c.2357+1G>A |
p.(?) |
5 |
PVS1, PM2, PP3, PS4 |
||||
|
NM_001005242.3 |
c.2446-2A>C |
p.(?) |
5 |
PVS1, PM2, PP3, |
||||
|
NM_004572.3 |
Deletion (exon 4) |
p.(?) |
4 |
PVS1, PM2 |
||||
|
DSC2 |
|
|
|
|
||||
|
NM_024422.6 |
c.923C>G |
p.Ser308fs* |
4 |
PVS1, PM2 |
||||
|
NM_024422.6 |
c.929delinsTT |
p.Gln310fs* |
4 |
PVS1, PM2 |
||||
|
NM_024422.6 |
c.1053_1059del |
p.His351Glnfs* |
4 |
PVS1, PM2 |
||||
|
DSG2 |
|
|
|
|
||||
|
NM_001943.5 |
c.523+1G>C |
p.(?) |
5 |
PVS1, PM2, PP3, PS4 |
||||
8 - Question: Are the nonsense variants in DSC2 and DSG2 heterozygous or homozygous? (Table 1). Recently, it was shown that also recessive DSC2 and DSG2 mutations contribute to ARVC/ACM. See 'A homozygous DSC2 deletion associated with arrhythmogenic cardiomyopathy is caused by uniparental isodisomy' and 'Hemi- and homozygous loss-of-function Mutations in DSG2 (Desmoglein-2) cause recessive arrhythmogenic cardiomyopathy with an early onset'. This should be explained and discussed.
Response: The following sentence was added to clarify this issue:
“All variants in PKP2, DSC2 and DSG2 of this cohort were in heterozygous. We did not have any recessive arrhythmogenic cardiomyopathy in the patients of this cohort.”
9- Question: Could you please insert a paragraph about the statistical analysis, which you used for your data analysis in the material and methods section?
Response: Thank you for your feedback; we have separated the paragraph to enhance its visual clarity:
Statistical analysis:
The SPSS statistical program version 21 (IBM Corporation, Armonk, New York) and R version 4.0 (R Core Team, 2021) were used for statistical analyses. Shapiro Wilk's normality test was applied for data representation and for subsequent statistical analyses Continuous data are presented as mean ± standard deviation (SD) or median {IQ range}. Fisher’s exact test and chi-square test were used for categorical data comparison and Wilcoxon or T-test for the comparison of continuous variables. Receiver operating characteristic (ROC) analysis was performed to determine optimal cutoff values of strain parameters and to obtain the sensitivity and specificity of each variable for detecting arrhythmic outcomes. A P value < 0.05 was considered significant.
Reviewer 2 Report
Comments and Suggestions for Authors
This paper investigates whether whether these advanced measures of right ventricular (RV) dysfunction 20 on echocardiogram increase diagnostic value for ARVC disease detection and to evaluate the association of echocardiographic parameters with arrhythmic outcomes. There are some concerns about this study.
- The motivation of this study should be further explained. In the literature, there have some similar studies. So what is the (potential clinical value) brought by this study ?
- It is better to present the inclusion and exlusion criterions in data collection.
- More studies on echocardiography image analysis should be cited to enhance the literature review, e.g.: Fully automatic segmentation of LV from echocardiography images and calculation of ejection fraction using deep learning; Multi-level semantic adaptation for few-shot segmentation on cardiac image sequences; Echocardiography segmentation with enforced temporal consistency.
- Please briefly indicate how the p-values are computed.
- More contents should be added in the captions of figures and tables for clarification.
- The study limitations should be enriched by adding more explanations.
- The key conclusion of this study should be highlighted.
- Some grammatical errors.
Comments on the Quality of English LanguageModerate refinement
Author Response
Response to Review 2
Thanks for the time you dedicated to the review. Your insightful feedback and constructive suggestions were valuable to our work. Thank you for your time and consideration.
Our point-by-point response are below:
1- Question: The motivation of this study should be further explained. In the literature, there have some similar studies. So what is the (potential clinical value) brought by this study ?
Response: In fact, only few previous studies have explored advanced echocardiographic techniques in patients with arrhythmogenic cardiomyopathy, and these studies were analyzed from slightly different perspectives. The implementation of new technologies in echocardiography occurred after the establishment of diagnostic criteria by the Task Force in 2010. Therefore, these data should be applied to diverse populations to be considered in diagnostic updates. To better clarify this issue, we added the sentence as follow:
“The implementation of new technologies in echocardiography occurred after the establishment of diagnostic criteria by the Task Force in 2010. Therefore, these data should be applied to diverse populations to be considered in diagnostic updates.”
2- Question: It is better to present the inclusion and exclusion criterions in data collection.
Response: Thanks for the advice. We added the inclusion/exclusion criteria, as follows:
“The inclusion criteria were patients who met the diagnostic criteria for arrhythmogenic right ventricular cardiomyopathy (according to the TFC 2010) and who signed the Informed Consent Form. The exclusion criteria were patients diagnosed with other specific cardiomyopathies, such as Chagas cardiomyopathy, sarcoidosis, and ischemic cardiomyopathy.”
3- Question: More studies on echocardiography image analysis should be cited to enhance the literature review, e.g.: Fully automatic segmentation of LV from echocardiography images and calculation of ejection fraction using deep learning; Multi-level semantic adaptation for few-shot segmentation on cardiac image sequences; Echocardiography segmentation with enforced temporal consistency.
Response: Thanks for this enriching suggestion. We added a paragraph with future directions, as follow:
“We believe that for the future, the anatomical, geometric, functional and segmental analysis of ventricles and atria could take into account automatic evaluations based on deep learning algorithms, multi-level semantic adaptation and temporal consistency models, bringing less interobserver variation in diverse analyzes leading to a shorter study time (39-43).”
4- Question: Please briefly indicate how the p-values are computed.
Response: In Table 3 (ECHO Parameters in ARVC) the p-values were computed by Wilcoxon test or T-test depending on previous normality test. Wilcoxon test was applied for non-normal variables and the T-test for normal variables. In Table 4 (Arrhythmic outcomes and STE parameters) the p-value was computed by chi-square test. We added an indication of each statistical test in the legend of the tables.
5- Question: More contents should be added in the captions of figures and tables for clarification.
Response: We added more content in the captions of the tables and figures.
“Table 1- The cohort was composed of young patients, most man and with an arrhythmic clinic phenotype.”
Table 2- Pathogenic or likely pathogenic variants demonstrating codon and protein alterations and the ACMG classification. The pathogenic variants were predominant in PKP2 gene. ACMG: American College of Medical Genetics.
Table 3: Echocardiogram parameters according to right and left ventricle parameters. The cohort has a right ventricle dilation and dysfunction predominance.
Table 4: Right and left atrial and ventricle parameters according to arrhythmic outcomes: right ventricle dilation was associated with arrhythmic outcomes.
Figure 1- Representative image of right ventricular parameters measurement. A) Four-chamber echocardiographic view with measures of strain. Right ventricle free wall longitudinal strain and right ventricle global longitudinal strain demonstrating RV dysfunction: -15.2% and -12.8% values, respectively. B) Tissue Doppler echocardiographic pulsed-wave showing RV peak systolic velocity (s’ wave) of tricuspid annulus in 10.1 cm/s. C) Four-chamber echocardiographic apical view with measures of RV diastolic area change, RV systolic area change, and fractional area change estimated at 37 cm², 29.3 cm², and 20.8%, respectively. RVFWSL: Right ventricle free wall strain longitudinal. RV4CSL: Right ventricle four-chamber strain.
Figure 2- Echocardiographic measurements reveal variations in values based on arrhythmic outcomes, indicating a discernible trend toward increased dilation and dysfunction in patients experiencing arrhythmic events.
6- Question: The study limitations should be enriched by adding more explanations.
Response: We added more explanations about study limitations, as follow:
“This study has several limitations. First, it is a single referral center, so a potential referral bias with overestimated severity result in a cohort of patients with a more severe phenotype. Second, we reported data from a relatively small population. Although ARVC is a rare disease, the sample size of the cohort was relatively modest, which might influence the generalizability of our results. A larger and more diverse participant pool would enhance the external validity of our conclusions. Third, specificity was not predicted to be studied because a control group was not analyzed. Long-term reassessments with second analyses of echocardiogram over time would be particularly beneficial in capturing potential changes in echocardiographic measurements and arrhythmic outcomes that may not be immediately evident. The echocardiographic measurements are subject to variability in interpretation and interobserver and intraobserver variability may influence the accuracy of the results. Lastly, the absence of certain clinical covariates in our analysis may limit the depth of interpretation of the results. Despite the limitations above, our study recognizes that ECHO is an imaging tool of important value and that these advanced acquisitions data provide information on ECHO’s usefulness to identify ARVC abnormalities that could improve ARVC detection.”
7- Question: The key conclusion of this study should be highlighted.
Response: We highlighted the key conclusion, as follow:
“In summary, the study highlights the nuanced characteristics of RV abnormalities in ARVC and suggests potential windows of opportunity for advancing diagnostic tools and risk assessment strategies in the future.”
8- Question: Some grammatical errors.
Response: We corrected grammatical errors and typing, we added some prepositions, coma and made some modifications to enhance clarity of the sentences. These changes and corrections are signalized in red in the manuscript.

Round 2
Reviewer 1 Report
Comments and Suggestions for Authors
Congratulations. The authors have improved their manuscript according to the suggested points. Therefore I suggest to accept this manuscript for publication.
Reviewer 2 Report
Comments and Suggestions for Authors
No further question.
Comments on the Quality of English LanguageMinor revision